

# Pests, diseases and crop protection practices in the smallholder sweetpotato production system of the highlands of Papua New Guinea

Geoff M. Gurr[1,2,3], Jian Liu[1,2], Anne C. Johnson[3], Deane N. Woruba[4], Gunnar Kirchhof[5], Ryosuke Fujinuma[5], William Sirabis[6], Yapo Jeffery[6] and Ramakrishna Akkinapally[7]

[1] State Key Laboratory of Ecological Pest Control for Fujian and Taiwan Crops, Fujian Agriculture & Forestry University, Fuzhou, Fujian, China
[2] Institute of Applied Ecology, Fujian Agriculture & Forestry University, Fuzhou, Fujian, China
[3] Graham Centre for Agricultural Innovation, Charles Sturt University, Orange, New South Wales, Australia
[4] Elizabeth Macarthur Agricultural Institute, NSW Department of Primary Industries, Menangle, New South Wales, Australia
[5] School of Agriculture and Food Sciences, The University of Queensland, St Lucia, Queensland, Australia
[6] Highlands Regional Centre, National Agricultural Research Institute, Aiyura, Eastern Highlands Province, Papua New Guinea
[7] National Agricultural Research Institute, Lae, Morobe Province, Papua New Guinea

Corresponding author
Geoff M. Gurr, ggurr@csu.edu.au

## ABSTRACT

Sweetpotato (*Ipomea batatans*) is a food crop of global significance. The storage roots and foliage of crop are attacked by a wide range of pests and diseases. Whilst these are generally well controlled in developed countries using approaches such as clean planting material and monitoring with pheromone traps to guide insecticide use, research into methods suitable for developing countries has lagged. In Papua New Guinea (PNG), sweetpotato is grown extensively as a subsistence crop and commercial production as a cash crop is developing. We report results from a survey of 33 smallholder producers located in the Highlands of PNG where the crop is of particular importance. Surveys of interviewees' crops showed high levels of pest and disease impact to foliage, stems and storage roots, especially in crops that were several years old. Weevils (Curculionidae) were reportedly the most damaging pests and scab (caused by the fungus *Elisnoe batatus*) the most damaging disease. Most producers reported root damage from the former and foliar damage from the latter but the general level of knowledge of pest and disease types was low. Despite the apparency of pest and disease signs and symptoms and recognition of their importance by farmers, a large majority of producers reported practiced no active pest or disease management. This was despite low numbers of farmers reporting use of traditional cultural practices including phytosanitary measures and insecticidal plants that had the scope for far wider use. Only one respondent reported use of insecticide though pesticides were available in nearby cities. This low level of pest and disease management in most cases, likely due to paucity in biological and technical knowledge among growers, hampers efforts to establish food security and constrains the development of sweetpotato as a cash crop.

# INTRODUCTION

Among globally important food crops, sweetpotato (*Ipomoea batatus*) ranks number seven (*Clark et al., 2013*) but has been the subject of far less research than other staples such as potato (*Solanum tuberosum*) and wheat (*Triticum aestivum*) (*Clark et al., 2013*). This reflects the fact sweetpotato is a relatively minor crop in most developed countries in contrast to its widespread production in many tropical and sub-tropical, developing regions such as Africa, southern Asia and the Pacific where it is important for local consumption in subsistence communities (*Woolfe, 1992*; *Bourke, 2009*; *Loebenstein & Thottapilly, 2009*; *Zhang et al., 2009*). In these areas, sweetpotato is critical for food security as it is often a major source of calories as well as vitamins such as carotenoids which are vital in preventing malnutrition in children (*Lebot, 2010*; *Woolfe, 1992*; *Kismul, Van denBroeck & Lunde, 2014*).

The storage roots of sweetpotato have high sugar and water content making them highly susceptible to biotic threats, especially during storage and if roots have been damaged by harvesting or pest attack (*Woolfe, 1992*). In developed country production systems, losses are prevented by the availability of infrastructure such as cool storage facilities and rapid transportation systems. In subsistence production systems, however, post-harvest losses are avoided only by progressive harvest on-demand for immediate use (*Okonya et al., 2014*), with the general lack of infrastructure otherwise leading to high levels of damage (*Johnson & Gurr, 2016*). This slows the development of commercial production and the livelihood benefits that value chains and processing potentially offer to impoverished rural communities.

Sweetpotato is attacked by around 300 species of arthropods (*Talekar, 1991*) that can cause severe to complete crop loss, as well as at least 30 diseases (*Clark et al., 2013*; *Johnson & Gurr, 2016*) provide a recent, comprehensive review of those most common in smallholder production. The fact that sweetpotato is vegetatively propagated, either by storage root fragments (slips) or by stem cuttings means that there is high scope for transfer of pest and pathogen inoculum from old to new crops. For example, eggs and larvae of the sweetpotato weevil *Cylas formicarius* (Fabricius), an especially important pest, can be found in these propagules (*Hartemink et al., 2000*). Still more difficult for subsistence farmers to manage is the fact that plant pathogen inocula, especially of viruses, is readily multiplied and distributed in slips and cuttings (*Clark et al., 2012*). Pests and diseases of sweetpotato are generally well controlled in developed countries by the use of pathogen-tested (clean) planting material, pheromone trapping and pesticides (*Clark et al., 2013*; *Jansson & Raman, 1991*). In developing countries, however, these technologies are less available, particularly in outlying areas, and often unaffordable, making subsistence growers more reliant on traditional practices. These cultural practices include 'slash and burn' production in which crops are established on newly-cleared land. However, population growth and associated

land shortage makes it increasingly difficult to continue these cultural practices resulting in more intense production with shorter fallow periods (*Bourke, 2001*). A further factor that exacerbates the potential impact of pests and diseases in developing countries is that sweetpotato is often grown in a small production unit (garden) as a series of consecutive crops for multiple years rather than as an annual cop rotated among multiple fields as in developed countries. This increases the time period over which pest densities and pathogen inoculum and infection levels can reach damaging levels, potentially compounded by depletion of nutrients from the soil resulting from repeated harvest of storage roots (*Bailey, 2009*; *Hughes et al., 2009*; *Kirchhof et al., 2009*).

Overall, sweetpotato production in developing countries is critical for food security but threatened—in a general sense—by pests and diseases, and effective management is difficult because well-studied technologies that are used in developed counties are not appropriate. Further, traditional practices that have allowed production for many generations are becoming less viable because of land shortages whilst research on management approaches that can be implemented has lagged because these regions are often lack funding and capacity for agricultural research. To address this situation, the aim of this study was to capture data that would identify the major biotic threats to sweetpotato production as a guide to future investment of research funding. The geographical focus of the study was the Highlands of PNG where this crop is the main food staple and where there are currently efforts to establish sweetpotato as a commercial cash crop. Whilst agronomic and soil management issues in this region have been the subject of some earlier research (*Kirchhof et al., 2009*; *Wegener, Kirchhof & Wilson , 2009*), no information has been available on pests, diseases and their management. A group of the authors visited 33 farmers spanning the major sweetpotato growing areas of the Highlands, conducting an extended interview with each and collecting data from their crops. Retail outlets in the two major towns of the region were also visited to determine the availability of pesticides.

## METHODS

Sweetpotato farmer surveys were conducted in the Highlands region of PNG in 2014 covering the same sites used in a 2005 survey of farming systems and soil management (*Kirchhof et al., 2009*; *Wegener, Kirchhof & Wilson, 2009*). The survey covered the five population centres of Asaro and Lufa in the Eastern Highlands Province, Gumine and Sinasina in the Simbu Province, and Mount Hagen in the Western Highlands Province. The Highlands region experiences sporadic outbreaks of inter-tribal conflicts and armed violence is common. Significant areas of potentially productive land sited between population centres is either uncultivated or is being overgrown with revegetation because it is considered too dangerous for people to regularly cultivate. Reflecting these hazards, local officials, village extension workers and police were used to facilitate an initial visit to population centres for the purposes of this study. Armed police accompanied the research team for one centre. At each centre, a preliminary meeting was held with the community in which authors able to speak the local dialect explained the nature of the survey and sought their participation. Thereafter, six to seven farmers from each village were surveyed,

a total of 33. Conditions did not permit detailed assessments and replicated destructive sampling for each site so the survey consisted of a rapid rural appraisal (RRA) (*Kirchhof et al., 2009*). Responses of interviewees were recorded on a standardised form in English. Interviewees were then asked to take the research team (4–5 persons depending on date) to a representative 'new garden' in which few successive sweetpotato crops had been grown and a representative 'old garden' in which many successive sweetpotato crops had been grown and that was planned to be placed into fallow or planted to a non-sweetpotato crop in the near future. Gardens of both categories were made available on most sites. A total of 27 local varieties were reported from these gardens with I Don't Care (7), Wahgi Besta (4), Susan's Black Eye (2) and Carrot Kaukau (2) being the only varieties present in more than one garden. All gardens were well established and producing storage roots at the time of inspection. Yield data for the sites and the region in general are not available because sweetpotato is grown as a subsistence crop that is harvested in a progressive manner. Gardens varied in size from approximately 50 m$^2$ to 200 m$^2$. This small size allowed the whole garden to be visually assessed for presence/absence of foliar symptoms. Permission was sought to harvest two randomly-selected sweetpotato plants from each garden. This was granted in a majority of cases (more readily for old than newly-planted gardens). The base of the stems was split to assess incidence of weevil larvae and their feeding tunnels and all of the storage roots beneath sampled plants were inspected for the presence of holes smaller than 3 mm in diameter and holes with greater diameter. Chi square analyses using the Quantpsy tool (*Preacher, 2001*) were used to compare old and new gardens, and compare the distribution of farmer responses within garden ages.

Concurrent with the farmer survey, the senior author visited all rural supply retailers in the major townships in the region, Goroka and Mount Hagen, to determine the availability of insecticide and fungicide products.

## RESULTS

The 33 farmers made available for inspection a total of 27 newly planted gardens and 28 old gardens.

### Crop inspections

The incidence of crops that were free of foliar symptoms was significantly ($P < 0.05$) lower for old than new gardens (Fig. 1). Deformities of the young leaves symptomatic of scab disease, caused by the fungus *Elisnoe batatus* Viégas & Jenkins, were the most common symptoms in old and new gardens. This was distinct from more general stunting of leaf size and discolouration (including mosaic) characteristic of viral diseases which was observed as frequently as scab symptoms in the old gardens. Viral symptoms were significantly ($P < 0.05$) less frequently observed among new than old gardens (Fig. 1). Splitting stem bases of sweetpotato vines was possible for only some gardens because growers tended to be concerned about destructive inspection of even a single plant because of the small size of the gardens but growers more inclined to approve this in old gardens (Fig. 2). Weevil larvae were detected in six of the 14 old gardens but only one of the 10 new gardens, however the small sample size meant that this difference was not significantly different

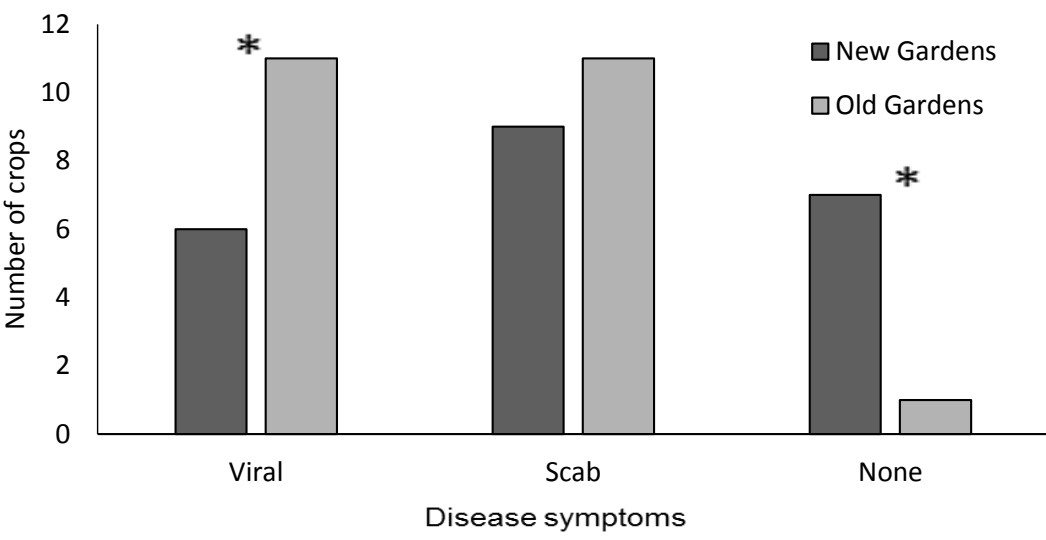

**Figure 1** **Incidence of foliar symptoms (viral infection and scab infection and symptom-free) among sweetpotato crops ($n = 25$ new and 20 old).** Symptoms were non-mutually, exclusive; some crops had symptoms of more than one type. (Chi-square tests compared old and new gardens: viral, $X^2 = 4.543$, $df = 1$, $p = 0.033$; scab, $X^2 = 1.635$, $df = 1$, $p = 0.202$; no symptoms, $X^2 = 4.021$, $df = 1$, $p = 0.044$).

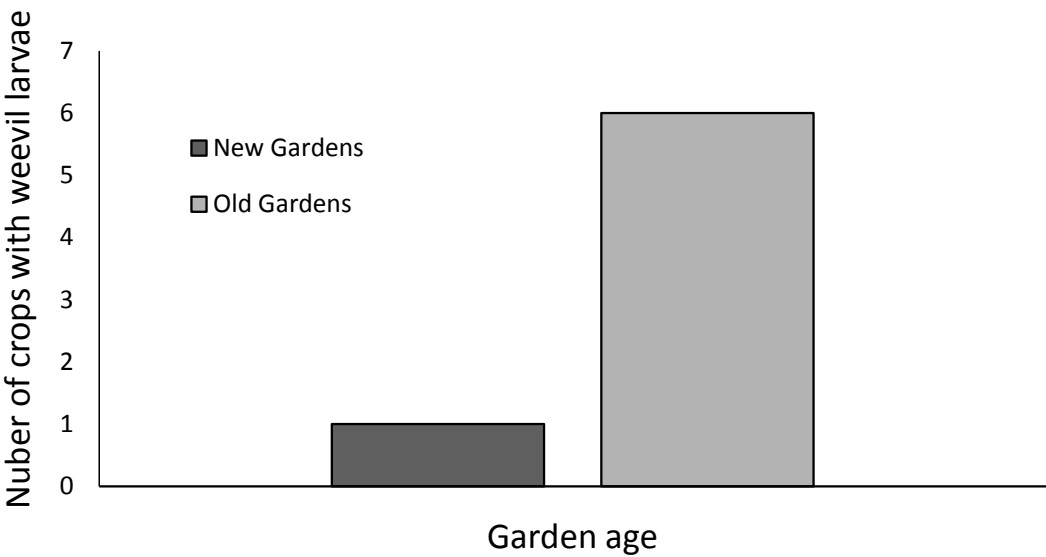

**Figure 2** **Incidence of weevils in the base of the stems among sweetpotato crops ($n = 10$ new and 14 old).** (Chi-square test compared old and new gardens $X^2 = 3.048$, $df = 1$, $p = 0.081$).

($P > 0.05$) (Fig. 2). For gardens of both ages, crops in which holes were consistently absent from all storage roots sampled from both randomly selected plants were in the minority (Fig. 3). The storage roots in most of the old gardens had small (<3 mm diameter) holes typical of sweetpotato weevil *C. formicarius*. Larger (>3 mm diameter) holes that may have been caused by the gregariously-feeding West Indian sweetpotato weevil *Euscepes postfasciatus* (Fairmaire) as well as other pests such as molluscs and rats was less common
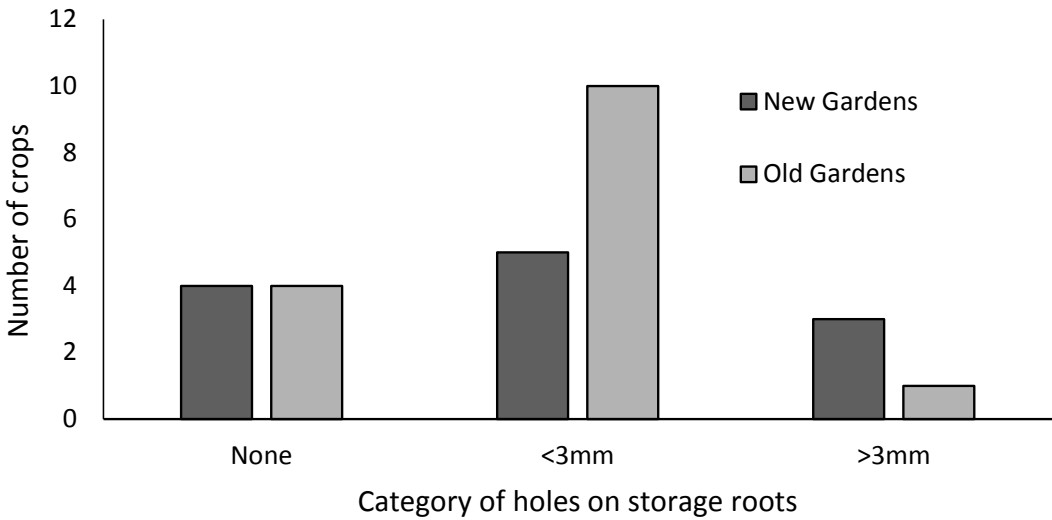

**Figure 3** **Incidence of pest damage holes in sweetpotato storage roots ($n = 10$ new and 14 old).** (One crop had holes of both sizes.) (Chi-square tests compared old and new gardens: <3 mm: $X^2 = 1.143$, $df = 1$, $p = 0.285$; >3 mm: $X^2 = 2.194$, $df = 1$, $p = 0.138$)

than smaller holes for gardens of both ages. For neither category of hole did the incidence differ significantly between old and new gardens.

## Farmer responses

New gardens reportedly had an average of 2.9 successive plantings (including the current crop) with an average fallow period between crops of 11.40 months compared with 25.8 successive plantings for old gardens with just 2.45 months between crops. Prior to the establishment of these gardens, the new ones had an average of 7.56 years of fallow with responses as high as "more than 50 years," whilst the old gardens were in fallow for 7.39 years with responses extending to "too long ago to remember". Farmers' expectation of storage root yield were most commonly high for new gardens and low for old gardens with differences between garden ages very highly significant ($P < 0.001$) (Fig. 4).

Very low number of farmers reported that their crops tended not to be attacked by pests and diseases (Fig. 5). Damage from these biotic factors was very much the norm. Chi square analysis comparing the null hypothesis of uniform pest attack across all plant parts with the farmers' reports of which plant parts were attacked showed significant ($p < 0.05$) differences for new gardens such that storage roots (the harvestable portion) were most attacked and roots least attacked (Fig. 5A). The same trend across plant parts was apparent among old gardens but the distribution of pest attack did not differ significantly from the null hypothesis. For diseases, stems and leaves were reportedly most commonly attacked and roots least attacked, a trend that was consistent across both garden ages and significantly different from the null hypothesis ($p < 0.05$) within each age (Fig. 5B). Caterpillars were considered a particular problem at the 5–6 month stage and gall mites and scab at harvest time.

Sweetpotato weevil (species unspecified) was ranked by the farmers as the crop protection issue of greatest concern and for which they most wanted a solution. Chi square analysis

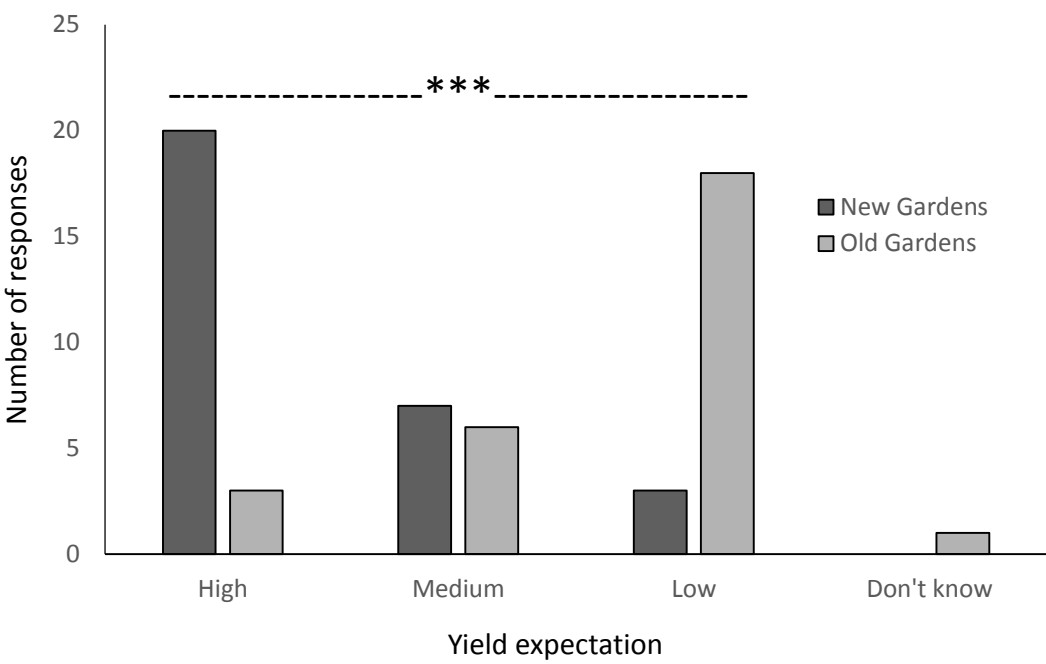

**Figure 4 Yield expectation of farmers for new and old gardens.** (Chi-square test compared distribution of responses between garden ages: $X^2 = 24.316$, $df = 3$, $p < 0.001$).

comparing the null hypothesis of all pest types reported with equal frequency with the farmers' reports showed very highly significant ($p < 0.001$) differences within new and old gardens (Fig. 6). This applied to the extent that weevils ranked more highly than all other biotic threat responses combined. Gall mite was the second highest ranked pest priority for gardens of both ages whilst grasshoppers and crickets were also specific concerns. Scab was the highest-ranked sweetpotato disease problem, again in gardens of both ages. 'Nematode,' 'tuber rot,' 'rust' and other, unknown diseases were also mentioned as biotic issues of concern. When asked to specify the times of year pest were most problematic the responses were varied. For sweetpotato weevil, attack was reported by farmers at widely varying times of the year and plant growth period but was mostly associated with the dry season. For crickets, planting and wet seasons were periods of reported risk. Gall mites and scab were of greatest concern at harvest time.

Despite all farmers noticing pests and diseases (Fig. 5) and considering pest damage, particularly by weevils, as a concern (Fig. 6) very few reported taking action to prevent or control pest attack. The great majority of farmers reported taking no action to manage pests (Fig. 7A). Chi square analysis comparing the null hypothesis of all pest management approaches (including no control) being reported with equal frequency with the farmers' reports showed very highly significant ($p < 0.001$) differences for new and old gardens (Fig. 7A). No more than four farmers each used the soil management approaches of mounding-up over storage roots or breaking up mounds to expose roots to heat; biological control with ants or chickens, mulching with plant materials such as 'fish-kill' (*Tephrosia* spp.) or other insecticidal plants. One farmer mentioned use of insecticide,
a

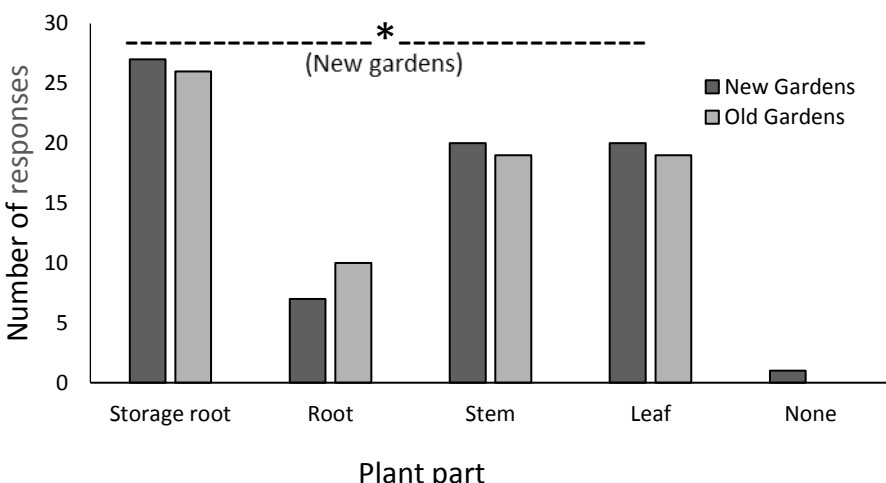

b

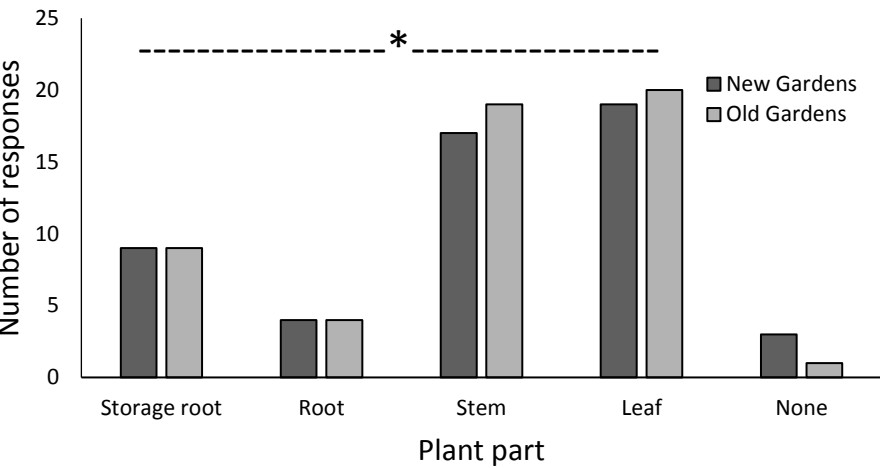

**Figure 5** **Farmers' responses on whether and where they observe damage by pests (A) and diseases (B).** (Means are the number of farmers mentioning a given concern and are non-mutually exclusive, some farmers mentioning one, and some multiple plant portions.) (Chi-square tests compared plant portions within each garden age: PESTS, new gardens, $X^2 = 7.849$, $df = 3$, $p = 0.049$; old gardens, $X^2 = 4.524$, $df = 3$, $p = 0.201$). DISEASES, old gardens: $X^2 = 8.544$, $df = 3$ $p = 0.036$; old gardens, $X^2 = 9.9444$, $df = 3$, $p = 0.0190$

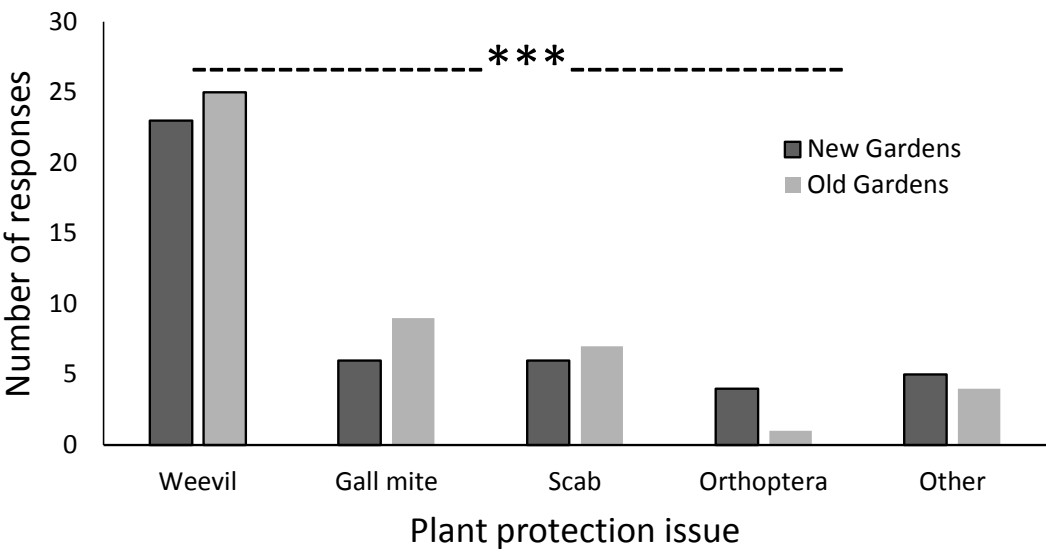

**Figure 6** **Plant protection issues cited in the top three concerns by farmers' for pest and disease problems.** Means are the number of farmers mentioning a given concern and are non-mutually exclusive, some farmers mentioning one, and some up to three issues. (Chi-square test compared pest types within each garden age: new gardens, priority is used by times been listed without giving any points. Weevil: $X^2 = 16.448$, $df = 4$, $p = 0.002$; old gardens, $X^2 = 23.836$, $df = 4$, $p < 0.001$).

Karate® (lambda-cyhalothrin) in his new garden. Only one grower reported the use of a combination of methods, soil management with rogueing (removal of infested stems), for pest management.

An equivalent lack of intervention was evident for disease management (Fig. 7B). Chi square analysis comparing the null hypothesis of all disease management approaches (including no control) being reported with equal frequency with the farmers' reports showed very highly significant ($p < 0.001$) differences for new and old gardens (Fig. 7B). One grower reported the use of 'clean planting material' but this was sourced from their own gardens rather than from a pathogen-tested planting material scheme. In a separate question specifically about use of planting material that was 'certified or disease tested,' all farmers reported no such use. One grower each reported rogueing (removal of symptomatic stems), fallowing and use of an unspecified resistant variety.

## Survey of pesticide availability

A survey of the seven rural supply shops in the two major townships of Mount Hagen and Goroka found that a small range of pesticides was available (Table 1). Of the eight insecticides available, only lambda-cyhalothrin was sold in most shops. Chlorothalonil was the only fungicide available in the two cities but on sale in most of the shops. Retailers reported these were usually purchased for use on cash crops such as Irish potato (*Solanum tuberosum*), allowing the cost of the input to be recouped, and rarely for use in sweetpotato since this was principally for consumption by the extended family. In some stores, the pesticides were repackaged into smaller, unlabelled packs for sale at low prices. More

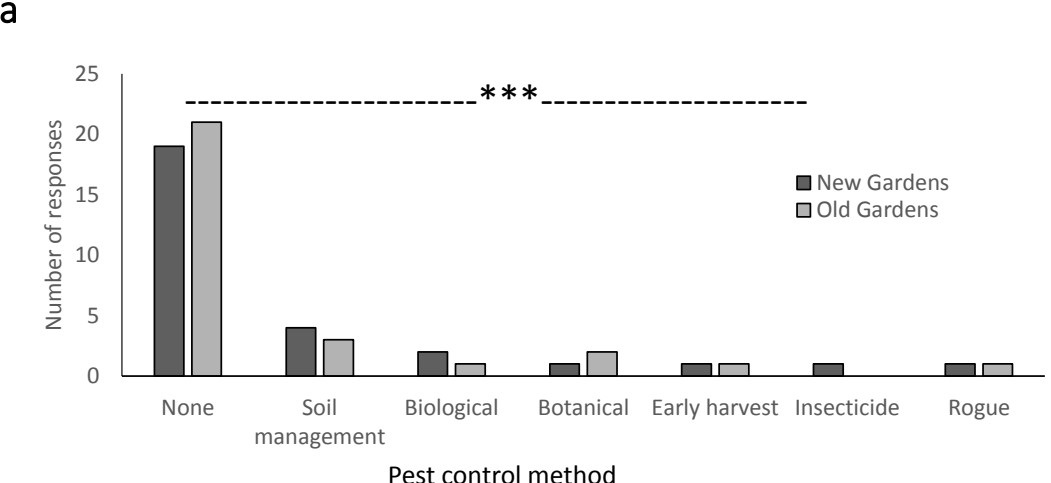

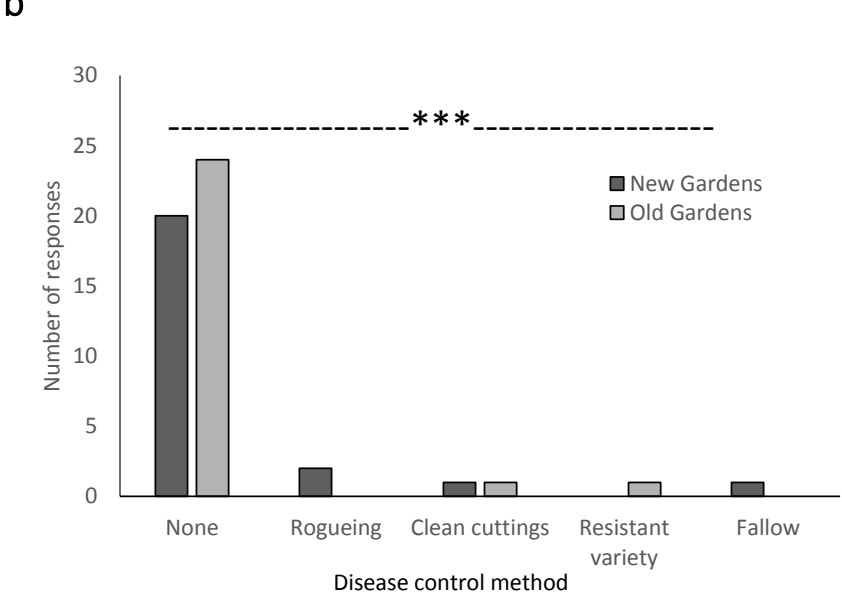

**Figure 7** **Reported actions taken to control pests (A) and diseases (B) on sweetpotato crops.** (Chi-square test compared management approaches within each garden age: PESTS, new gardens, $X^2 = 41.989$, $df = 6$, $p < 0.001$; old gardens, $X^2 = 52.738$, $df = 6$, $p < 0.001$. DISEASES, new gardens, $X^2 = 38.338$, $df = 4$, $p < 0.001$; old gardens, $X^2 = 52.277$, $df = 4$, $p < 0.001$).

generally, labelling practices were not stringent, with packs of one chlorothalonil product carrying the contradictory wording 'protective fungicide' and 'group Y herbicide' (Fig. 8).

## DISCUSSION

Developing country pest and disease issues tend to receive less attention than those in developed countries and this is compounded in regions where studies are made more
**Table 1** **Insecticide and fungicide availability in retail outlets in the Papua New Guinea Highlands region townships of Goroka and Mount Hagen.** (The anonymity of the retail suppliers is protected by de-identification and the use of lettering).

| City | Retail supplier | Product name and active constituent | Type |
|---|---|---|---|
| Goroka | A | Karate® 25 g/L; lambda-cyhalothrin | Insecticide |
| | | Eko® 720 g/L; chlorothalonil | Fungicide |
| | | Barrek® 500 g/L; chlorothalonil | Fungicide |
| | B | Lambda® C2.5EC; lambda-cyhalothrin | Insecticide |
| | | Malathion® ; malathion | Insecticide |
| | | Eko® 720 g/L; chlorothalonil | Fungicide |
| | C | Permethrin® 250 EC; permethrin | Insecticide |
| | | Lambda® C2.5EC; lambda-cyhalothrin | Insecticide |
| | | Bifenthrin® 10%; bifenthrin | Insecticide |
| | | Eko® 720 g/L; chlorothalonil | Fungicide |
| | D | Confidor® ; imidacloprid | Insecticide |
| Mount Hagen | E | Permethrin® ; permethrin | Insecticide |
| | | Carbofuran® ; carbofuran | Insecticide |
| | | Acephate® 75%wv; acephate | Insecticide |
| | | Bifenthrin® ; bifenthrin | Insecticide |
| | | Chlorpyrifos® 480EC; chlorpyrifos | Insecticide |
| | | Barrek® 500 g/L; chlorothalonil | Fungicide |
| | F | Eko® 720 g/L; chlorothalonil | Fungicide |

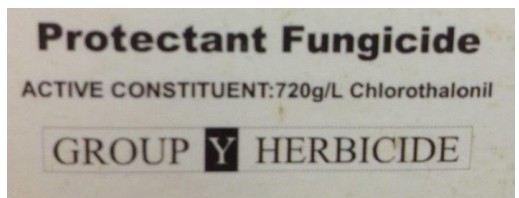

**Figure 8** **Example of pesticide labelling anomaly.** Photograph from pesticide label on product for sale in Goroka.

difficult because of instability and violence. Thus, though agricultural research in PNG has been the subject of significant effort in recent years, there is a relative dearth of information to inform priorities and investment. The present study of smallholder sweetpotato growers in the region of PNG, where this crop is the main staple, provides strong evidence that pests and diseases are having a large impact on production and that current management efforts are inadequate.

Among the biotic threats that farmers reported to be of high concern, weevils were paramount. This was evident also in the assessment of damage to storage roots and inspections of stems in which weevils were frequently present. Internationally, the sweetpotato weevil is consistently ranked as the most problematic pest in sweetpotato production (*Ebregt et al., 2004*; *Fielding & Van Crowder, 1995*; *Nsibande & McGeoch, 1999*; *Okonya et al., 2014*; *Parr, Ntonifor & Jackai , 2014*; *Placide et al., 2015*) though the damage

can be confused with that from other pests such as millipedes (Diplopoda) (*Ebregt et al., 2004*). *Euscepes postfasciatus* is present in PNG (*Hughes, 2013*) and this causes some forms of damage similar to that of the sweetpotato weevil (*C. formicarius*). Though the adults of these two weevils are dissimilar in appearance, the immatures look very similar. No farmers mentioned either species specifically so the relative importance of these two species as pests remains to be determined. Certainly, both are potentially serious pests. Weevil attack was reported by farmers at widely varying times of the year but was mostly associated with the dry season, reflecting the fact that storage roots are more exposed to attack if soil cracks as a result of dry conditions (*Lutulele, 2001*; *Parr, Ntonifor & Jackai , 2014*) and this suggests that impact could be more severe under climate change conditions (*Okonya et al., 2014*).

Native to the Indian subcontinent and eastwards to Malaysia, *C. formicarius* is a serious pest in the south west Pacific, the southern USA, Caribbean and South America (*Chalfant et al., 1990*; *Sherman & Tamashiro , 1954*; *Waterhouse & Norris, 1987*). *Austin, Jansson & Wolfe (1991)* and *Horton & Ewell (1991)* considered this pest of great importance in causing pre-harvest damage. *Euscepes postfasciatus* originated from the Caribbean and is now a pest in the Pacific region and South America (*Katsuki et al., 2012*; *Raman & Alleyne, 1991*; *Sherman & Tamashiro , 1954*). An important mode of dispersal for both species is as immatures within storage roots or stem cuttings (*Hartemink et al., 2000*; *Ray, Mishra & Mishra, 1983*). Larvae of both weevil species feed on the storage root or within stems causing tunnelling packed with frass. Adult *E. postfasciatus* tend to feed on storage roots gregariously, causing relatively few large holes. In contrast *C. formicarius* adults tend to feed individually causing smaller wounds (*Sherman & Tamashiro , 1954*). Accordingly, our classification of observed holes on storage roots into <3 mm and >3 mm diameter provides an approximate indication that *C. formicarius* may be the dominant weevil species. Clearly storage root holes could also be caused by other pests, such as molluscs and rats, especially in the case of larger holes, so these results are tentative. Studies based on rearing-out adults from infested storage roots or identifying immatures (potentially aided by the development of molecular diagnostic tools) are necessary in order to discriminate the incidence and impact of these two weevil species and plan appropriate research and management priorities and such studies are currently underway.

Gall mite, *Eriophyes gastrotrichus* Nalepa (Acari: Eriophyidae), causes erinose, a foliar disease characterised by blister-like galls on the stems of sweetpotato plants in the Philippines, and PNG where it is has previously been reported to be a problem of increasing concern in the Highlands (*Ames et al., 1996*; *Hughes et al., 2009*). This pest was the second most highly-rated concern among growers. Since it infests the foliage, it is readily spread by stem cuttings which are commonly used in the region. The use of slips or, more especially, pathogen-tested planting material would allow crops to be established in a 'clean' state and allow production for some time before field infection occurs. The Australian Centre for International Agricultural Research has invested in establishing a pathogen-tested planting material program in the region. Whilst the principal focus of this is control of viruses (see below) it would also benefit crop protection more widely including for gall mite. In the present study, however, none of the farmers reported prior use of planting material that was pathogen tested, certified or disease tested. Some reported use of 'clean planting

material' but this was sourced from their own or nearby gardens and illustrates that they were aware of this infection pathway and the need to manage carryover of inoculum.

Symptoms of scab, caused by *Elsinoe batatas* Viégas & Jenkins, was the most commonly observed form of foliar symptoms in both old and new gardens and was also the disease considered of highest priority by farmers. Though the symptoms of this disease are characteristic and unlikely to be confused with those of other diseases, pathogen isolation in future studies is required to confirm identity. Throughout tropical regions, scab is considered the most serious fungal disease of sweetpotato (*Clark et al., 2013*; *Coleman et al., 2009*). Though the storage roots can be infected this tends to cause little impact; though foliar damage can be so severe that photosynthetic area is reduced leading to storage root yield reductions as high as 34% (*Coleman et al., 2009*). Pathogen inoculum survives on crop residues and can be transmitted readily by stem cuttings so is chiefly a problem when sweetpotato is grown continuously (*Clark et al., 2013*; *Coleman et al., 2009*). It is noteworthy, then, that its incidence was high even in the new crop gardens and this reflects the fact that no farmers had accessed pathogen-tested clean planting material.

Viruses are widely considered to be of great economic importance in sweetpotato production (*Clark et al., 2012*; *Gibson & Kreuze, 2014*). A survey of scientists from less developed countries rated viruses as the top priority (*Fuglie, 2007*). Notably, however, no farmers in the present study mentioned viruses though a large proportion of old gardens showed foliar symptoms consistent with viral infection. As noted above, pathogen identification is required in future work to confirm the precise cause of these symptoms. The apparent lack of concern amongst growers about viral diseases likely reflects the fact that symptoms of viral infection can be subtle and develop over a prolonged period with little or no direct symptoms on the storage roots other than yield decline which is likely to be attributed to pests because of their greater apparency. Related to this, the concept of a plant pathogenic virus, that has no signs, is relatively unfamiliar to many farmers so it not being mentioned is likely to reflect this fact. The availability of molecular detection methods has led to rapid advances in sweetpotato virus knowledge and at least 30 viruses of sweetpotato are known (*Clark et al., 2012*), some with multiple strains (*Dolores, Yebron Jr & Laurena, 2012*). Yields of virus-infected sweetpotato plants are often severely affected, reduced by as much as 80–90% (*Carey et al., 1999*; *Clark et al., 2012*; *Davis & Ruabete, 2010*). Though insects such as aphids such as *Aphis gossypii* and whiteflies including *Bemisia tabaci* can transmit viruses (*Clark et al., 2012*; *Byamukama et al., 2004*), propagation material is the chief means of viral spread (*Gibson et al., 1997*; *Moyer & Larsen, 1991*; *Mbanzibwa et al., 2014*). Foliar symptoms of virus infection include leaf distortion, strapping and crinkling, mosaics, vein clearing, brown blotches and general stunting and chlorosis (*Mbanzibwa et al., 2014*). These symptoms were significantly more frequently seen in old rather than new gardens, reflecting the time available for plant-to-plant transmission and build up of infection levels.

These differences in pest and disease apparency between old and new gardens underscore the importance of political action to establish peaceful rural communities in order to allow potentially productive farmlands to be used. Prior to the establishment of these gardens, the new ones had 7.56 years of fallow whilst the old gardens were in fallow for 7.39 year,
less than half as long as the 16.8 (SE = 2.4) year reported for a 2005 survey of the same sites (*Kirchhof et al., 2009*). This shortening of fallows reflects land shortages resulting from rapidly increasing human population densities (*Bourke, 2001*) and is likely to allow pest and disease pressure to increase because fallowing has been demonstrated to increase yields via benefits to crop nutrition (*Hartemink, 2003*; *Hartemink et al., 2000*). Accordingly, if farming communities in the Highlands of New Guinea felt sufficiently safe to extend their cropping activities back into areas that has fallen out of production because of fear of inter-tribal violence, this would alleviate both biotic and abiotic (nutritional) stress on crops.

A striking finding about pest and disease management practices among the surveyed growers is the very large majority who reported not practicing any active management. This is despite the existence of a potentially large number of methods that could be employed in this setting. Small numbers of farmers reported using insecticidal plants, basic phytosanitation methods and simple forms of biological control using ants or livestock. The makum system is a traditional PNG practice for production of taro on mounds in which the ant, *Pheidole megacephala* (Fabricius), has nested, and has been adapted for use in sweetpotato production (*Sar et al., 2009*). Ants are also employed in a system in Cuba involving green tree ants being transported into sweetpotato fields from banana plantations within their rolled banana leaf nests (*Lagnaoui et al., 2000*). Ants can provide sweetpotato weevil control in a more cost effective than insecticides (*Chalfant et al., 1990*), so merits more attention as a method that could be readily adopted in smallholder systems. It is not possible to determine from the present study why such low rates of pest and disease management were apparent in the present study but the most likely explanation—based on general interactions with the farming communities—is lack of knowledge. In particular, though farmers recognised a range of pest and disease types, their knowledge of lifecycles and essential concepts such as microscopic disease causing agents was rudimentary. Further, though expectations of storage tuber yield from old gardens was lower than from new gardens, there was a tendency to associate this with nutrient depletion. Associated with this, the adoption of strategies to manage nutrition, such as not burning crop residues (*Bailey, 2009*), could exacerbate carryover of pests and pathogen inocula.

A survey of sweetpotato growers in Tanzania found that although farmers could identify diseased plants they could not distinguish the different types of disease (*Adam, Sindi & Badstue, 2015*). Though those African farmers had a very limited knowledge of pests and pathogens, they took at active precautions to manage them (*Adam, Sindi & Badstue, 2015*; *Nsibande & McGeoch, 1999*). For example, they identified plants that looked healthy and free of pests for use in planting material (*Adam, Sindi & Badstue, 2015*). This was not widely reported as a pest or disease management practice in the present survey though farmers are likely to select relatively healthy cuttings on the basis of these being likely to root readily and grow vigorously. The closer a village was to a main town or main road with passing traffic the more likely the farmers in the Tanzanian study were to be able to identify diseases that affect sweetpotato (*Adam, Sindi & Badstue, 2015*). Sites with easier access also tended to facilitate the use of higher quality planting material. In the present study, all sites were accessible by roads (*Kirchhof et al., 2009*) it is likely that levels of knowledge and

active pest and disease management are still lower in the more remote areas of the PNG Highlands. Farmer-to-farmer interactions are an important source of information sharing on pest management (*Adam, Sindi & Badstue, 2015*; *Pouratashi & Iravani, 2012*) but this communication channel is impeded in PNG by tribal conflict, and this underscores the importance of extension efforts and initiatives such as the development of a pathogen-tested planting material scheme. Among the challenges for such a scheme is that many dozens of sweetpotato varieties are grown in the Highlands of PNG so the scheme would need to 'clean-up' and make available a wide range of cultivars to meet farmers' needs.

## CONCLUSION

Like many developing countries, PNG is experiencing rapid population growth and government policies are seeking to establish greater food security and livelihood development, the latter by developing cash crops and value adding to agricultural commodities by processing and marketing. Sweetpotato potentially can contribute strongly to both these objectives because it is widely grown and culturally integral to traditional diets, yet strongly impacted by pests and diseases that are not well managed. The recent *IPES-Food (2016)* 'Uniformity to Diversity' Report highlighted the multiple negative outcomes from intensive agriculture in developed countries. These include loss of biodiversity and reliance on non-renewable and environmentally hazardous inputs including pesticides. Accordingly, the development trajectory of countries such as PNG need to be cognizant of the negative aspects of simply following practices already established in developed nation agricultural systems. For example, making pesticides more readily available and promoting their use are not logical from the sustainability perspective and would also complicate the common practice of feeding sweetpotato foliage to pigs. Production needs to be increased to meet human needs but achieving this by becoming reliant on non-renewable inputs and eroding the natural resource base of agriculture will lead to unsustainability (*Godfray & Charles, 2011*). As an alternative, ecological intensification (in which ecosystem services such as biological pest control and nutrient cycling are key) offers viable benefits (*Bommarco, Kleijn & Potts, 2013*). If wider use of pesticides is to be avoided, the need for alternative approaches is clear but traditional practices of ancient agricultural systems, such as ants and livestock for biological control, and insecticidal plants, can underpin this if their efficacy and utility are better understood and appropriate extension efforts are made. Parallel with such technological efforts, however, advances are necessary in the political and policy arena to make rural communities safer and more sustainable. Recent human population growth and inter-tribal conflict over ever-more-scarce land has resulted in more intensive cropping in areas close to villages exacerbating pest and disease build-up.

## ACKNOWLEDGEMENTS

Mr. Kai Lali (NARI) drove and provided field support. Sargent Simon Wakala (PNG Royal Constabulary) assisted with security.

### Funding

This work was supported as a 'Small Research Activity' by the Australian Centre for International Agricultural Research (SMCN/2012/016). Geoff Gurr was supported by a Chinese Government Thousand Talents Program fellowship during which this manuscript was prepared. The funders had no role in study design, data collection and analysis, decision to publish, or preparation of the manuscript.

### Grant Disclosures

The following grant information was disclosed by the authors:
Australian Centre for International Agricultural Research: SMCN/2012/016.
Chinese Government Thousand Talents Program.

### Competing Interests

Geoff Gurr is an Academic Editor for PeerJ. Geoff Gurr and Jian Liu are married. Geoff Gurr, Gunnar Kirchhof, Ryosuke Fujinuma and William Sirabis are funded by the Australian Centre for International Agricultural Research to conduct research arising in part from the results presented herein.

### Author Contributions

- Geoff M. Gurr conceived and designed the experiments, performed the experiments, analyzed the data, wrote the paper, prepared figures and/or tables, reviewed drafts of the paper.
- Jian Liu analyzed the data, wrote the paper, prepared figures and/or tables, reviewed drafts of the paper.
- Anne C. Johnson wrote the paper, prepared figures and/or tables, reviewed drafts of the paper.
- Deane N. Woruba, William Sirabis and Yapo Jeffery performed the experiments, reviewed drafts of the paper.
- Gunnar Kirchhof and Ramakrishna Akkinapally conceived and designed the experiments, reviewed drafts of the paper.
- Ryosuke Fujinuma conceived and designed the experiments, performed the experiments, reviewed drafts of the paper.

### Data Availability

Gaur P. 2016. Supplementary File. Figshare: 10.6084/m9.figshare.3838038.v1.

### Supplemental Information

Supplemental information for this article can be found online at http://dx.doi.org/10.7717/peerj.2703#supplemental-information.

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
