# Peer review of "Pests, diseases and crop protection practices in the smallholder sweetpotato production system of the highlands of Papua New Guinea"

_PeerJ, doi:10.7717/peerj.2703_

## Round 0.1 · original submission · Minor Revisions

Three reviewers appreciated the quality of your work. Please address their minor revisions.

·

Basic reporting

The manuscript entitled "Pests, diseases and crop protection practices in the smallholder sweet potato production system of the highlands of Papua New Guinea" provides important data about on plant disease and herbivore insects affecting sweet potato on sweet potato based on extensive farm survey. This paper was well thought out, and it is generally well written and the results and discussion are clearly presented. I could find no serious flaws in logic or presentation.

Experimental design

This paper was well thought out, and it is generally well written and the results and discussion are clearly presented. I could find no serious flaws in logic or presentation.

Validity of the findings

I would suggest to provide on figure 1-7 SE for each bar. And on table 1, I would suggest to provide the meaning of the letters used to identify the retail supplier.

·

Basic reporting

The ms is written in smooth language and its structure is formatted in accordance with Peer J style. There are minor lettering errors in figure1, 2 and 3 (figure 1, asterisk; figure 2, nube. figure 3; horizontal title). The raw data are available.

Experimental design

The study investigated the major pests and diseases as well as farmers protection practices for sweetpotato production in the highland of PNG. The research aim is well defined. In the introduction authors provided sufficient literatures to explain the importance of research. The survey is a combination of symptom observation and farmers interview using a small sample of 33 farmers. The description of methods is simple. There is no information such as the demography of survey location, planted sweetpotato variety and growth stage and annual yield, surveyed farmers’ educational levels. It is hard to evaluate the representativeness of survey.

Validity of the findings

The results are clearly stated. Significant differences were determined using the chi-square test. There are no data present about the major diseases incidence and severity. In addition, disease symptom alone is often inadequate for disease identification, why authors did not further isolate causal agents to confirm the major diseases in laboratory.

·

Basic reporting

This manuscript reports on a very basic survey of sweet potato insect pests and plant diseases in PNG, and also of crop protection options available. It is a very simple survey, but yields some interesting information. The writing is clear, and the analyses straightforward. The hypotheses tested are rather simple, but serve the purpose of addressing the basic questions addressed by this study.

Experimental design

Very basic and determined to a large extent by difficult field conditions, but probably adequate for the type of data collected, and the questions being addressed.

Validity of the findings

The data are straightforward and the findings are similarly straightforward and clear.

Additional comments

The finding that failure to select 'clean' propagative material was prevalent is quite interesting - an interesting contrast with the African situation.

---

## Round 0.2 · accepted · Accept

The manuscript can be accepted now.